# Conformational States of the GDP- and GTP-Bound HRAS Affected by A59E and K117R: An Exploration from Gaussian Accelerated Molecular Dynamics

**DOI:** 10.3390/molecules29030645

**Published:** 2024-01-30

**Authors:** Zhiping Yu, Zhen Wang, Xiuzhen Cui, Zanxia Cao, Wanyunfei Zhang, Kunxiao Sun, Guodong Hu

**Affiliations:** 1Shandong Key Laboratory of Biophysics, Dezhou University, Dezhou 253023, China; yuzhiping900414@aliyun.com (Z.Y.); qiayilai@mail.ustc.edu.cn (Z.C.); 2Pingyin People’s Hospital, Jinan 250400, China; himkey@163.com (Z.W.); 15806617027@163.com (X.C.); 3School of Science, Xi’an Polytechnic University, Xi’an 710048, China; 210811059@stu.xpu.edu.cn (W.Z.); sunkunxiaoxpu@163.com (K.S.)

**Keywords:** HRAS, mutation, GaMD simulations, free energy landscapes, principal component analysis

## Abstract

The HRAS protein is considered a critical target for drug development in cancers. It is vital for effective drug development to understand the effects of mutations on the binding of GTP and GDP to HRAS. We conducted Gaussian accelerated molecular dynamics (GaMD) simulations and free energy landscape (FEL) calculations to investigate the impacts of two mutations (A59E and K117R) on GTP and GDP binding and the conformational states of the switch domain. Our findings demonstrate that these mutations not only modify the flexibility of the switch domains, but also affect the correlated motions of these domains. Furthermore, the mutations significantly disrupt the dynamic behavior of the switch domains, leading to a conformational change in HRAS. Additionally, these mutations significantly impact the switch domain’s interactions, including their hydrogen bonding with ligands and electrostatic interactions with magnesium ions. Since the switch domains are crucial for the binding of HRAS to effectors, any alterations in their interactions or conformational states will undoubtedly disrupt the activity of HRAS. This research provides valuable information for the design of drugs targeting HRAS.

## 1. Introduction

The GTPase proteins, specifically HRAS, KRAS, and NRAS, function as molecular switches that regulate intracellular signaling pathways in cell growth and differentiation [1,2,3]. These RAS proteins share a similar structure and can transition between an active, GTP-bound state and an inactive, GDP-bound state [4,5,6,7,8]. Numerous studies have shown that mutations in the three RAS proteins are implicated in various human cancers [9,10,11], highlighting their distinct roles in exploring potential cancer therapies [12,13]. Furthermore, these mutations significantly impact the conformational dynamics of RAS proteins, which play a critical role in their interactions with cell signaling proteins [14,15]. Given their importance in anticancer treatment, considerable attention has been directed towards designing small molecule inhibitors that target allosteric sites on RAS proteins. Additionally, comprehending the conformational dynamics of RAS proteins is crucial for inhibitor development. Therefore, uncovering the molecular mechanisms underlying mutation-induced conformational changes in RAS proteins is essential for advancing effective anti-cancer drugs.

Recently, there has been increased attention on HRAS, one of the three RAS proteins (HRAS, KRAS, and NRAS), due to its involvement in mutation-induced oncology activity [16,17,18,19,20,21,22,23,24]. HRAS has two switch domains, switch domain 1 (SW1) and switch domain 2 (SW2), as shown in Figure 1A. SW1 consists of residues 28–40, while SW2 is composed of residues 59–75. The switch domains SW1 and SW2 undergo significant changes due to GTP hydrolysis into GDP and point mutations [25,26,27,28]. Experimental structures have revealed that point mutations and ligand bindings cause conformational alterations in the switch domains of HRAS [28,29,30,31,32]. Johnson et al. employed X-ray crystallography, nuclear magnetic resonance spectroscopy, binding assays, and molecular dynamics (MD) simulations to investigate the autophosphorylated mutants of HRAS and KRAS. Their findings suggested that the phosphoryl transfer from GTP requires dynamic movement of the SW2 [33]. Matsumoto et al. determined the experimental structure of the HRAS T35S/GppNHp complex. They found that T35S not only caused noticeable conformational alterations in the switch domains, but also implicated the naphthalene ring as a potential scaffold for developing RAS inhibitors [34]. Matsumoto et al.’s study indicated that oncogenic mutations Q61L and Q61H may disrupt intramolecular structural communication between the switch domains SW1 and SW2 of RAS proteins [35]. Buhrman et al., using multiple solvent crystal structures and computational solvent mapping (FTMap), explored binding site hot spots in the “off” and “on” allosteric states of GTP-associated HRAS. Their findings suggested that the global conformational rearrangement of HRAS acts as a mechanism for allosteric coupling between the effector interface and remote hot spots in all RAS isoforms [36]. The aforementioned experimental structures provide valuable information for further investigating the function of HRAS and its potential role in future drug design.

Apart from experimental works, various computational technologies have been extensively utilized to elucidate the molecular mechanism of ligand–target identification and investigate ligand–HRAS binding modes. These technologies include MD simulations [37,38,39,40,41,42], Gaussian accelerated molecular dynamics (GaMD) simulations [43,44], calculations of binding free energies [45,46,47,48,49,50,51], and principal component analysis (PCA) [52,53,54,55]. The research conducted by Lu’s group involved MD simulations, binding free energy prediction, and dynamic network analysis, which suggested that the R135K mutation triggers the allosteric unbinding of monobody from HRAS [56]. Gorfe et al. employed MD simulations and PCA to quantitatively decipher the structural and dynamical features of the active and inactive states of GTP- and GDP-bound HRAS. Their results indicated that the inactive-to-active transformation of HRAS is a multiphase process involving the rearrangement of the switches SW1 and SW2 [57]. Furthermore, different research groups utilized accelerated molecular dynamics (aMD) simulations [58] to study the conformational changes in RAS proteins mediated by mutations [59,60]. Bao et al. employed GaMD simulations and FELs analysis to explore the alterations in the switch states of HRAS induced by point mutations D33E, A59T, and L120A and to clarify the impact of these mutations on the conformational dynamics of HRAS [61]. Although those experimental and computational studies provided valuable insights into the regulation of RAS activity, a comprehensive understanding of the conformational dynamics of HRAS and its function is still lacking. Therefore, further investigations are essential to exploring the conformational changes of HRAS caused by mutations to deepen our understanding of the function of HRAS.

To further investigate the molecular mechanism underlying conformational changes in HRAS caused by point mutations, two specific mutations, A59E [33] and K117R [27], were selected for analysis in this study. The mutated sites are illustrated in Figure 1B. The A59E mutation occurs near the SW2 region, while the K117R mutation is at the allosteric site. The substitution of A59 with E59 not only leads to an elongation of the sidechain, but also introduces a negative charge. On the other hand, the K117R mutation only alters the geometry of the sidechain. Compared to the frequently detected mutations of codons 12 and 13, A59E and K117R also lead to functional differences in HRAS. The study by Denayer et al. proposed that the K117R mutation disrupts guanine nucleotide binding and has similar functional implications as mutations affecting GTP hydrolysis and causing human diseases [27]. On the other hand, A59E is a mutation of RAS that is associated with cancer. The research conducted by Johnson et al. confirmed that both A59E and phosphorylation significantly accelerate the intrinsic exchange [33]. In addition, K117R has weaker effects on downstream c-Jun N-terminal kinase signaling. Thus, to explore the effects of these two mutations on the conformational changes in HRAS, we constructed six systems: wild-type (WT) HRAS bound to GDP, A59E HRAS bound to GDP, K117R HRAS bound to GDP, WT HRAS bound to GTP, A59E HRAS bound to GTP, and K117R HRAS bound to GTP. The structures of GDP and GTP are depicted in Figure 1C,D, respectively. As reported in previous studies, GaMD simulations have demonstrated great success in exploring the dynamics of target conformations [15,62,63,64,65,66,67]. Therefore, GaMD simulations were used in this study to enhance the conformational sampling of HRAS. Subsequently, post-processing analyses were performed on the GaMD trajectories, including principal component analysis (PCA), dynamics cross-correlation maps (DCCMs) [66,68,69,70,71], and free energy landscape (FEL) analyses. This work is anticipated to provide valuable insights and information to aid in future drug design targeting RAS proteins.

## 2. Results and Discussion

### 2.1. Structural Fluctuation and Flexibility

The RMSDs of heavy atoms from the two ligands, GTP and GDP, in the binding pocket of the WT and mutated HRAS were calculated relative to their initial optimized structure to understand their dynamics in HRAS (Appendix A). The RMSDs of GTP fluctuate between 0.16 Å and 3.12 Å in the active form of HRAS (Appendix A), while those of GDP range from 0.06 Å to 4.72 Å in the inactive state of HRAS (Appendix A). This result shows that the stability of GTP in active HRAS is higher than that of GDP in inactive HRAS. The frequency distribution of RMSDs for GTP and GDP are presented in Figure 2A,B, respectively. In WT HRAS, A59E, and K117R HRAS mutants (Figure 2A), the RMSDs of heavy atoms for GTP are distributed with peaks at 0.67 ± 0.01, 0.91 ± 0.01, and 0.74 ± 0.00 Å, and the average RMSDs are 0.72, 1.07, and 0.81 Å. Moreover, the averaged RMSDs of GTP in the A59E and K117R HRAS are 0.35 and 0.09 Å larger than those in the WT one. Thus, the A59E and K117R mutations slightly weaken the structural stability of GTP in HRAS compared to the WT HRAS, and the influence of A59E is stronger than that of K117R. As for GDP, the distribution of RMSDs has one peak of 0.91 ± 0.01 and 1.02 Å ± 0.01 in the WT and A59E HRAS, respectively, but two peaks of 1.08 ± 0.01 and 2.25 ± 0.02 Å for the K117R HRAS (Figure 2B). The average RMSDs are 1.08, 1.16, and 1.99 Å for the GDP-bound WT, A59E, and K117 HRAS, respectively. These results indicate that A59E and K117R mutations reduce the structural stability of GDP in HRAS compared to the WT HRAS, and the effect of K117R is more significant than that of A59E.

The RMSFs of the Cα atoms in HRAS were computed in order to understand the impacts of A59E and K117R on the structural flexibility of HRAS (Figure 2C,D). The structural domains corresponding to the alterations in the RMSFs are depicted in Appendix A. It can be observed that the SW1, SW2, and the loop L2 showed high flexibility (Figure 2C,D and Appendix A). The difference in RMSFs between the WT and mutated HRAS was estimated using the equation $∆RMSF={RMSF}_{mutant}-{RMSF}_{WT}$ (Appendix A), where ∆RMSF, ${RMSF}_{mutant}$ and ${RMSF}_{WT}$ indicate the RMSF difference and RMSFs of mutants and WT HRAS, respectively. In the case of GTP-bound WT and mutated HRAS, K117R strengthens the structural flexibility of the switch domain SW1 in the GTP-bound active HRAS by referencing the WT HRAS (Figure 2C); differently, A59E weakens the structural flexibility of the SW1-L (the loop part of the SW1) but strengthens that of the SW1-β, namely, the β part of the SW1 (Appendix A). A59E slightly enhances the structural flexibility of the switch domain SW2 in the GTP-bound active HRAS, but K117R highly reduces the structural flexibility of this domain compared to the GTP-bound WT one (Appendix A). In addition, A59E increases the structural flexibility of the P-loop, while K117R produces the opposite effect on the structural flexibility of the L2 loop. For the GDP-bound WT and mutated HRAS, although A59E enhances the structural flexibility of the α1-L compared to the WT HRAS, it weakens that of the SW1-β (Appendix A). K117R abates the structural flexibility of the α1-L, but highly strengthens that of the SW1-L and the loop L2 by comparison with the GDP-bound WT HRAS (Appendix A). Furthermore, K117R increases the structural flexibility of the P-loop, and residues 84–98 and A59E strengthen the structural flexibility of the L2 loop compared to the GDP-bound WT HRAS (Figure 2D).

The MSAs (molecular surface areas) of the GTP- and GDP-bound forms of HRAS were estimated to evaluate the extent to which HRAS is exposed to solvents (Appendix A). It is worth noting that the MSAs of the GTP- and GDP-bound WT, A59E, and K117R variants of HRAS exhibit similar fluctuating ranges and display stable fluctuations (Appendix A). The MSAs of the GTP-bound WT, A59E, and K117R HRAS are concentrated around 8142, 8597, and 8276 Å^2^, respectively, indicating that A59E and K117R mutations increase the exposure of HRAS to solvents compared to the GTP-bound WT HRAS (Appendix A). The MSAs of the GDP-bound A59E and K117R variants show a 310 Å^2^ increase compared to the GDP-bound WT HRAS (Appendix A), suggesting that these two mutations, A59E and K117R, also enhance the exposure of HRAS to solvents relative to the GDP-bound WT HRAS.

In summary, the mutations A59E and K117R significantly impact the structural stability of GTP and GDP within the binding pocket of HRAS, thus affecting their interaction with HRAS. Furthermore, A59E and K117R greatly influence the structural flexibility of the SW1, SW2, and L2 loop regions, which disrupt the binding of HRAS to effectors and the allosteric regulation of HRAS activity. Additionally, A59E and K117R increase the surface areas of the GTP-bound active form of HRAS and the GDP-bound inactive form that are exposed to solvents, affecting the HRAS activity. Previous studies have also shown the influence of these two mutations on the switch domains of HRAS, providing strong support for our findings [27,33].

### 2.2. Effect of A59E and K117R on Dynamics of HRAS

DCCMs offer valuable insight into the alterations in the internal dynamics of targets resulting from point mutations and ligand associations, which are crucial for drug design toward these targets. In this study, DCCMs were computed using the coordinates of the Cα atoms in HRAS to investigate the impact of mutations on the internal dynamics of HRAS (Figure 3 and Appendix A). Our findings demonstrate that A59E and K117R mutations lead to substantial disruptions in correlated motions between structural regions of the GTP-bound active and GDP-bound inactive forms of HRAS.

For the GTP- and GDP-bound WT HRAS (Figure 3A,B), the regions R1 and R2 show the anti-correlated motions of the P-loop relative to the switch domains SW1 and SW2, respectively. The region R3 describes the positively correlated motions of the loop L1 and the helix α3 relative to the P-loop. In contrast, the region R4 characterizes the anti-correlated movements of the loop L1 and the helix α3 relative to the SW2 (Figure 1A and Figure 3A,B). In addition, the region R5 shows the anti-correlated motion between the L3 loop and the SW2 (Figure 1A and Figure 3A,B). Compared to the GTP-bound WT HRAS, A59E weakens the anti-correlated motion between the P-loop and the SW1, but K117R strengthens the anti-correlated motion of this region (Figure 3C,E and Appendix A). K117R not only leads to the disappearance of the anti-correlated motion of the P-loop relative to the SW2, but also induces the appearance of positively correlated motion between the P-loop and the SW2 by referencing the GTP-bound WT HRAS (Figure 3C,E and Appendix A). On the contrary, A59E hardly affects the anti-correlated movement in the R2 or the positively correlated motion of the R3 in the GTP-bound active HRAS (Figure 3B). Differently, K119R weakens the positively correlated motion of the region R3 relative to the GTP-bound WT HRAS (Figure 3C and S5A). Although A59E and K117R scarcely affect the anti-correlated movement of the region R4, these two mutations reduce the anti-correlated motion of the region R5 compared to the GTP-bound WT HRAS (Figure 3C,E and Appendix A). Regarding the GDP-bound HRAS, K117R hardly changes the anti-correlated movements of the P-loop relative to SW1 and SW2 or the positively correlated motion of the region R3 by comparison with the GDP-bound WT HRAS (Figure 3D,F and Appendix A). However, A59E highly weakens the anti-correlated movements of the P-loop relative to the SW1 and SW2. It slightly reduces the positively correlated motion of the region R3 by referencing the GDP-bound WT HRAS (Figure 3D,F and Appendix A). By comparison with the GDP-bound WT HRAS, A59E significantly weakens the anti-correlated movement of the region R4 (Figure 3D and Appendix A), and K117R slightly abates the anti-correlated motion of this region (Figure 3D,F and Appendix A). In addition, A59E and K117R almost lead to the disappearance of anti-correlated movement in the region R5 relative to the GDP-bound WT HRAS (Figure 3D,F and Appendix A). In the regions above, the R1 and R2 correspond to the switch domains SW1 and SW2, respectively, while the R3, R4, and R5 are involved in allosteric regulation sites on the activity of HRAS. Thus, the changes in the internal dynamics of the regions R1–R5 caused by A59E and K117R can significantly impact the binding of HRAS to its effectors and allosteric regulations on the activity of HRAS.

With the expectation of deeply probing the mutation-mediated effect on the dynamics behavior of HRAS, PCA was carried out through diagonalization of a covariance matrix constructed using the coordinates of the Cα atoms. The function of eigenvalues vs. eigenvectors arising from PCA was calculated (Appendix A). The results suggest that A59E and K117R strengthen structural fluctuation along the first eigenvector relative to the GTP-bound WT HRAS (Appendix A). In the GDP-bound inactive HRAS, K117R remarkably enhances the structural fluctuation of HRAS along the first eigenvector compared to the GDP-bound WT HRAS, but A59E reduces the structural fluctuation of HRAS along the first eigenvector (Appendix A). The above analyses imply that these two mutations possibly affect the activity of HRAS.

To further explore the effects of A59E and K117R on the dynamics behavior of HRAS, the first eigenvector resulting from the PCA is visualized in Figure 4. GTP. The GDP and mutation sites are also depicted in Figure 4. The results indicate that two switch domains SW1 and SW2 in HRAS show high structural flexibility and that mutations heavily affect their motion behavior. Compared to the GTP-bound WT HRAS (Figure 4A), A59E not only leads to more disordered states of the SW1, as indicated by unordered arrow directions, but also inhibits the fluctuation amplitude of the SW2 along the first eigenvector (Figure 4C). Meanwhile, A59E changes the motion direction of the L3 loop (Figure 4C). By referencing the GTP-bound WT HRAS (Figure 4A), K117R not only results in the SW2 moving in the opposite direction, but also strengthens the fluctuation amplitude of the SW1 along the first eigenvector (Figure 4E). In addition, K117R also alters the movement direction of the L3 loop relative to the GTP-bound WT HRAS (Figure 4E). By comparison with the GDP-bound WT HRAS (Figure 4B), A59E completely changes the relative motion direction of the helix α3 to the SW2. Meanwhile, it obviously weakens the fluctuation of the SW2 and α3, which possibly generates an effect on the allosteric regulation (Figure 4D). Furthermore, A59E also changes the motion direction of the L3 loop and the SW1 compared to the GDP-bound WT HRAS (Figure 4D). In the GDP-bound K117R HRAS, the motion direction of the helix α3 and the loop L3 are hardly changed due to K117R relative to the GDP-bound WT HRAS, and their fluctuation amplitudes are slightly weakened because of K117R (Figure 4F). More interestingly, the motion directions of the SW1 and SW2 are altered by K117R; furthermore, the fluctuation amplitudes of these switch domains are also decreased compared to the GDP-bound WT HRAS. Thus, A59E and K117R significantly influence the dynamics behavior of the SW1, SW2, L3, and α3 in the GTP- and GDP-bound HRAS.

Based on the analyses described above, A59E and K117R alter the internal dynamics of the GTP-bound active and GDP-bound inactive HRAS, particularly the switch domains. It is well known that the switch domains SW1 and SW2 are involved in the binding of HRAS to its effectors. Hence, the changes in dynamics behavior of the SW1 and SW2 caused by A59E and K117R affect the HRAS–effector binding and regulate the activity of HRAS, which is supported by the later distance analyses in the FELs. In addition, the two current mutations also disturb the dynamic behavior of the loop L3 and the helix α3, which are involved in the allosteric position of HRAS. Therefore, A59E, and K117R certainly influence the allosteric regulation of HRAS activity. G12V, Q61L, and the other mutations can significantly change the conformational dynamics of the SW1 in HRAS and impact the binding of HRAS to effectors [61,72,73], which generously supports our current findings.

### 2.3. Effect of Mutations on Free Energy Profiles of HRAS

It is well known that free energy profiles can provide significant information to deeply understand the influences of A59E and K117R on conformational changes in the GTP- and GDP-bound HRAS. We constructed free energy landscapes (FELs) using two reaction coordinates: the distance between the Cα atoms of residue D33 in the SW1 and residue E62 in the SW2, as well as the RMSDs of heavy atoms from HRAS (Figure 5). The distance can effectively capture conformational transitions between SW1 and SW2, while the RMSDs accurately represent the overall structural fluctuations of HRAS. The reason we chose them as reaction coordinates is precisely because of this.

In the GTP-bound WT HRAS, GaMD simulations captured two energy basins, EB1 and EB2 (Figure 5A, left panel). The EB1 basin was deeper than EB2, indicating that the conformational transition from EB2 to EB1 in the GTP-bound WT HRAS is more accessible than the reverse transition. The conformations at EB1 and EB2 accounted for 81.2% and 17.8% of the total conformations, respectively (Figure 5A, left panel). Therefore, the GTP-bound WT HRAS conformations were mostly found in the EB1. Two representative structures falling into EB1 and EB2 were aligned (Figure 5A, left panel) to reveal structural differences. The SW1 from the GTP-WT HRAS complex exhibited significant deviations between the EB1 and EB2, whereas the remaining structures of the GTP-bound WT HRAS were aligned very well (Figure 5A, right panel). The distances from D33 to E62 were 15.6 and 26.0 Å in the EB1 and EB2 states, respectively, in the GTP-associated WT HRAS. These findings suggest that the GTP-bound WT HRAS switch domains are located at a tighter state in the EB1 than in the EB2. For the GTP-bound A59E HRAS, only one energy basin, EB1, was detected by GaMD simulations (Figure 5B, left panel), and the representative structure of the GTP-bound A59E located in the EB1 was extracted from the GaMD trajectory (Figure 5B, middle panel). Compared to the GTP-bound WT HRAS, A59E led to a less energetic state, implying that A59E induces conformational rearrangement of HRAS. At this state, the distance between D33 and E62 was 18.2 Å (Figure 5B, right panel), which is different from the switch state of the GTP-bound WT HRAS. For the GTP-bound K117R HRAS, GaMD simulations revealed the presence of two primary energy basins, EB1 and EB2 (Figure 5C, left panel), with EB1 being significantly more profound than EB2. Consequently, the transition probability from the EB2 to the EB1 was much higher than the reverse transition from the EB1 to the EB2. The conformations located at EB1 and EB2 accounted for 88.3% and 10.3% of the total structures, respectively, indicating that most of the conformations of the GTP-bound K117R HRAS are distributed in the EB1 basin. The two representative structures located at EB1 and EB2 were superimposed, as shown in Figure 5C (middle and left panels). The results indicate that the K117R mutation caused a noticeable deviation of the loop L2 compared to the GTP-bound WT HRAS. Interestingly, the loop L2 was located near the allosteric position of HRAS, suggesting that K117R may affect the allosteric regulation of HRAS activity. In the EB1 state of the GTP-bound K117R HRAS, the distance between D33 and E62 was 16.2 Å, while in the EB2 state, it was 28.9 Å (Figure 5C, middle and left panels). Therefore, the switch domains of the GTP-bound K117R HRAS are looser than those of the GTP-bound WT HRAS, indicating that the K117R mutation influences the binding of HRAS to its effectors.

Regarding the GDP-associated WT HRAS, GaMD simulations identified two energy basins, EB1 and EB2 (Figure 5D, left panel). The conformations found in EB1 and EB2 accounted for 63.4% and 34.6% of the total saved conformations from the GaMD trajectories, respectively (Figure 5D, left panel). The structural changes were examined by superimposing two representative structures from EB1 and EB2. The results suggest that the switch domains SW1 and SW2 exhibited significant deviations between the two energetic states EB1 and EB2 of the GDP-bound WT HRAS. In contrast, the other domains maintained good alignment (Figure 5D, middle panel). In both energy states EB1 and EB2, the distances between D33 and E62 in the GDP-bound WT HRAS were 15.9 Å (Figure 5D, right panel). Although the relative position between D33 and E62 remained unchanged, the helix α2 of the SW2 was closer to the helix α3 in the EB2 compared to the EB1 (Figure 5D, right panel). In the case of the GDP-bound A59E HRAS, two energy basins, EB1 and EB2, were observed (Figure 5D, left panel). The depth of these energy basins indicated a higher transition probability from the EB2 to the EB1 than that from the EB1 to the EB2. The conformations located in the EB1 and EB2 accounted for 74.2% and 24.2% of the total conformations, respectively, suggesting that most of the GDP-bound A59E HRAS conformed in EB1 (Figure 5D, right panel). To examine the structural differences, two representative structures from EB1 and EB2 were aligned. The results revealed distinct deviations in the SW1 between the two energy states, while the SW2 in the GDP-bound A59E HRAS appeared to be more ordered than the GDP-bound WT HRAS (Figure 5E, left panel). The distances between D33 and E62 in the EB1 and EB2 states of the GDP-bound A59E HRAS were 16.5 Å and 26.8 Å, respectively (Figure 5E, right panel). This finding indicates that the A59E mutation results in a looser switch domain compared to the GDP-bound WT HRAS, providing evidence for the impact of A59E on the binding of HRAS to its effector. As for the GDP-bound K117R HRAS, GaMD simulations also revealed the presence of two energy basins, EB1 and EB2 (Figure 5F, left panel). The depth of the EB1 was found to be greater than that of the EB2 in the GDP-bound K117R HRAS. It was observed that 63.4% of the total conformations saved in the GaMD trajectories were trapped in the EB1. In comparison, 34.6% were observed in the EB2, indicating that the majority of conformations of the GTP-bound K117R HRAS were predominantly distributed in the EB1. A comparison of the EB1 and EB2 superimposed structures, representative of the GDP-associated HRAS (Figure 5F, middle panel), revealed significant deviations in the SW1 and the loop L3, which may potentially impact HRAS-effector binding and allosteric regulation. The distances between D33 and E62 in the GDP-bound K117R HRAS located in the EB1 and EB2 were measured to be 18.3 Å and 18.2 Å, respectively (Figure 5F, right panel), a considerable increase compared to the corresponding distances in the GDP-bound WT HRAS. These findings suggest that the K117R mutation induces a looser switch domain than the WT HRAS, which likely affects the binding between HRAS and its effector.

Based on the analyses conducted, it is evident that the two mutations, A59E and K119R, significantly impact the conformational states of HRAS, particularly in the switch domains. These mutations disrupt the compactness of the switch domains in HRAS, which are crucial for HRAS–effector binding. Consequently, the mutations interfere with the binding of HRAS to its effectors and disturb its normal function. Furthermore, the mutations also cause changes in the relative positioning of α2 and α3 compared to the GDP-bound WT HRAS. Since α3 is involved in regions responsible for allosteric regulation, the two mutations can potentially influence the allosteric regulation of HRAS activity. Previous studies have also demonstrated that point mutations, such as T35S, G12D, and G12C, can disrupt HRAS activity by altering the conformational dynamics of the switch domains [59,74,75]. These findings align with the results obtained in our research.

### 2.4. Analyses of GTP- and GDP-HRAS Interaction Networks

The interaction networks between GTP and GDP with HRAS were analyzed using the protein–ligand interaction profiler (PLIP) server [76] (Figure 6). The hydrogen bonding interactions (HBIs) of GTP and GDP with HRAS were also identified using the CPPTRAJ program (Table 1; Figure 6A,B). We investigated the role of Mg^2+^ in the binding of ligands to HRAS (Figure 6C,D). The frequency distributions of the distances involved in the non-bonded interactions of GTP and GDP with residues are shown in Figure 7, and the time courses of these distances are provided in Appendix A. The frequency distributions of the distances involved in the Mg^2+^-mediated interactions in the GTP- and GDP-bound systems are plotted in Figure 7, while the time evolution of these distances is displayed in Appendix A.

The residues in the P-loop, including G13, G15, K16, S17, and A18 in WT HRAS, formed HBIs with GTP and GDP (Figure 6A,B), respectively. These interactions had high occupancies, exceeding 60.0% (Table 1), indicating their high stability throughout the GaMD simulations of GTP- and GDP-bound WT HRAS. S17 formed a weaker HBI with GDP compared to GTP. In contrast to the GTP-associated WT HRAS, GDP formed an additional HBI with V14, but its occupancy was only 14.2%. Additionally, GTP and GDP also formed HBIs with the residues N116, K117, and D119 in loop L3, as well as residues S145, A146, and K147 in loop L4 of WT HRAS (Figure 6A,B). These HBIs, except for those involving K117, had occupancies larger than 52.8% and remained stable throughout the entire GaMD simulations of GTP- and GDP-bound WT HRAS (Table 1). The occupancies of four HBIs between GTP and residues V29, D30, Y32, and T35 from the switch region SW1 in WT HRAS exceeded 60.0%. In comparison, the occupancies of three HBIs between GDP and residues V29, D30, and Y32 ranged from 29.8% to 39.7% (Table 1), suggesting weaker HBIs between SW1 and GDP compared to GTP. When comparing to GTP- and GDP-bound WT HRAS, the A59E mutation resulted in a noticeable reduction in the occupancy of HBIs between GTP and residues V29, D30, Y32, T35, and D119, as well as a decrease in HBIs between GDP and residues G13, G15, A18, V29, D30, Y32, D119, S145, and K117. However, A59E significantly increased the occupancy of HBIs between GDP and V14 and S17 (Table 1). These findings indicate that A59E had a more significant impact on HBIs between GDP and HRAS than between GTP and HRAS. On the other hand, the K117R mutation substantially decreased the occupancy of HBIs between GTP and residues G13, S17, V29, D30, Y32, T35, N116, D119, and K147. It also reduced the occupancy of HBIs between GDP and residues G13, K16, A18, V29, D30, Y32, D119, and A146. Furthermore, K117R abolished the HBIs of GTP and GDP with residue 117. Changes in the stability of HBIs between GTP, GDP, and HRAS caused by A59E and K117R can impact the activity of HRAS.

Based on Figure 6A, GTP formed two salt bridge interactions with K16, a π–π interaction with F28, and a salt bridge interaction with D119. On the other hand, GDP only formed a salt bridge interaction with K16 while maintaining the same interactions with F28 and D119 as those observed in GTP (Figure 6B). The distances involved in the salt bridge interactions of GTP and GDP were calculated using the GaMD trajectories. The time courses and frequency distributions of these distances are illustrated in Figure 7 and Appendix A. The temporal evolution of the distances between GTP and K16 for the two salt bridge interactions demonstrates that these salt bridges exhibited remarkable stability throughout the GaMD simulations. Specifically, in the GTP-bound WT, A59E, and K117R HRAS, the distances between the nitrogen atom NZ of K16 and the phosphorus atom PB of GTP were distributed at 3.35, 3.41, and 3.35 Å, respectively (Figure 7A). Furthermore, the consistent population of distances between the nitrogen atom NZ of K16 and the phosphorus atom PG at 3.71 Å (Figure 7A) suggests that the A59E and K117R variants had minimal impact on the stability of the two salt bridge interactions with K16. While the distances between the mass centers of the phenyl ring in F28 and the guanine group of GTP remained within a stable fluctuation range (Appendix A), their frequency distributions revealed that the GTP-bound A59E HRAS exhibited two peak values (Figure 7A). This indicates that K117R had little influence on the π–π interaction between GTP and F28, while A59E slightly weakened this interaction compared to the GTP-bound WT HRAS. The time evolution of the salt bridge interaction distance between GTP and D119 suggests that this distance remained highly stable (Appendix A), with both the GTP-bound WT and mutated HRAS exhibiting distributions centered around 3.45 Å. Consequently, A59E and K117R had minimal impact on the stability of this salt bridge. In contrast, the distance between the nitrogen atom NZ of K16 and the phosphorus atom PA showed significant fluctuation over the simulation time (Appendix A), with most of its frequency distribution located at 6.19 Å (Figure 7B). This indicates that the phosphorus atom PA of GDP did not form a salt bridge interaction with the nitrogen atom NZ of K16; instead, it only yielded electrostatic interactions. Furthermore, K117R slightly weakened this electrostatic interaction. On the other hand, the distance between the nitrogen atom NZ of K16 and the phosphorus atom PB of GDP exhibited a stable fluctuation (Figure 4B), with values of 3.53, 3.47, and 3.63 Å observed in the GDP-bound WT, A59E, and K117R HRAS, respectively (Figure 7B). This suggests that the nitrogen atom NZ of K16 formed a salt bridge with the phosphorus atom PB of GDP. Thus, K117R slightly disrupted the salt bridge interaction between GDP and K16 compared to the GDP-bound WT HRAS. According to Figure 7B, the phenyl ring of F28 engaged in a π–π interaction with the hydrophobic ring of GDP, and the distances involved in this interaction exhibited stable fluctuations (Appendix A). As depicted in Figure 7B, the distances between the mass centers of the phenyl group in F28 and the hydrophobic ring of GDP were approximately 5.11 Å in the GDP-bound WT and K117R HRAS variants. In contrast, the GDP-bound A59E HRAS exhibited two peak values and a wider distribution for this distance (Figure 7B). Consequently, K117R minimally influenced the π–π interaction between GDP and F28, while A59E weakened this interaction relative to the GDP-bound HRAS. The distances corresponding to the salt bridge interaction between GDP and D119 fluctuated within a stable range in both the GDP-bound WT and mutated HRAS (Appendix A), with their frequency distributions largely overlapping (Figure 7B). This suggests that A59E and K117R did not significantly affect the salt bridge interaction between GDP and D119.

To investigate the effects of mutations on the stability of Mg^2+^, we analyzed the distances of Mg^2+^-mediated interactions using the CPPTRAJ program. The results are presented in Figure 7 and Appendix A. The distances between Mg^2+^ and the oxygen atom OG of S17 exhibited stable fluctuations during the GaMD simulations of the GTP-bound WT, A59E, and K117R HRAS variants (Appendix A). The frequency distributions of these distances overlapped (Figure 7C), indicating that A59E and K117R had minimal impact on the interaction between Mg^2+^ and S17. In contrast, the distances between Mg^2+^ and the oxygen atom O1G of T35 fluctuated significantly during the GaMD simulations of all three systems (Appendix A). The frequency distributions suggest that A59E weakened the electrostatic interaction between Mg^2+^ and O1G compared to the GTP-bound WT HRAS, while K117R strengthened this interaction (Figure 7C). Furthermore, the distance between Mg^2+^ and the mass center of the oxygen atoms OD1 and OD2 of D57 was highly unstable during the GaMD simulations of the GTP-bound WT and mutated HRAS (Figure 7C and Appendix A). K117R significantly reduced the electrostatic interaction between Mg^2+^ and the carbonyl group of D57. The distances between Mg^2+^ and the phosphorus atoms PB and PG of GTP showed stable and narrow fluctuations throughout the simulations of the three systems (Appendix A). Their frequency distributions almost overlapped (Figure 7C), indicating that A59E and K117R had minimal impacts on the electrostatic interactions between Mg^2+^ and GTP. In Appendix A, it can be observed that the distances between Mg^2+^ and the oxygen atom OG of S17 were highly unstable during the simulations of the GDP-bound WT, A59E, and K117R HRAS. The GDP-bound WT HRAS frequency distribution display a single peak, while the GDP-bound A59E and K117R HRAS show two peaks. Furthermore, the second peak in the mutated HRAS is shifted to the right (Figure 7D), suggesting that A59E and K117R weakened the electrostatic interactions between Mg^2+^ and OG of S17. According to Appendix A, the distances of Mg^2+^ from the oxygen atom O1G of T35 in switch region SW1, as well as the mass center of the oxygen atoms OD1 and OD2 from D57 near switch region SW2, exhibited high unordered fluctuations (Appendix A), which is supported by their frequency distributions with multiple peaks (Figure 7D). The distances of Mg^2+^ from the oxygen atoms O1B and O2B of GDP frequently transitioned between two states during the GaMD simulations of three GDP-bound HRAS systems (Appendix A). The frequencies of these distances show opposite distributed states in the WT relative to the mutated HRAS (Figure 7D). Figure 7D indicates that A59E and K117R strengthened the electrostatic interactions of Mg^2+^ with O1B of GDP compared to the GDP-bound WT HRAS. At the same time, Figure 1D verifies that these two mutations weakened the electrostatic interactions of Mg^2+^ with O2B of GDP.

The aforementioned analyses show that A59E and K117R had significant impacts on the interaction networks involving ligands and Mg^2+^. (1) These two mutations significantly decreased the stability of hydrogen bonds between the ligands and the switch domains. (2) A59E and K117R disrupted the electrostatic interactions between Mg^2+^ and the residues T35 and D57 within the switch domains. (3) A59E and K117R affected the electrostatic interactions between Mg^2+^ and the phosphate group of GDP. It is well-established that the switch domains largely overlap with the binding domains of HRAS to effector proteins. Therefore, any alterations in the conformational states of the switch domains will undoubtedly influence the binding and disrupt the function of HRAS. These findings are consistent with previous studies [33,73,77], further validating our current research.

## 3. Theory and Methods

### 3.1. System Constructions

The crystal structure (PDB entry: 4EFL) of the GppNHp-bound WT HRAS, obtained from the Protein Data Bank (PDB) [73], was utilized as a template to replace the GTP analog GppNHp with authentic GTP, resulting in the generation of the GTP-bound WT HRAS. Meanwhile, A59 and K117 were substituted with E59 and R117, respectively, forming the GTP-bound A59E and GTP-bound K117R. The initial coordinates of the GDP-bound A59E HRAS were taken from the PDB, and its entry was 7JII [33]. The E59 in 7JII was mutated to A59 to form the GDP-bound WT HRAS. Subsequently, the residue K117 was replaced by R117 to derive the GDP-bound K117R HRAS. A magnesium ion Mg^2+^ was located at the same site in both crystallographic structures and remained in the constructed systems. The protonated states were checked and assigned to each residue with the program H++ 3.0 [78]. The missing hydrogen atoms in the crystal structures were added using the Leap module in the Amber 20 package [79,80]. The ff19SB force field [81] was employed to produce the force field parameters of the WT and mutated HRAS. The GDP and GTP force field parameters were obtained using the general Amber force field (GAFF2) [82,83]. The Austin Model 1 method with bond charge correction (AM1-BCC) [84] was adopted to derive the atomic charges of two ligands, GDP and GTP, through the Antechamber tool in Amber [85]. All systems were immersed in an octahedral, periodic box of water with a buffer of 10.0 Å to reflect the solvent environment, in which the TIP3P model [86] was utilized to yield the force field parameters of water molecules. The appropriate number of sodium (Na^+^) and chloride (Cl^−^) ions were added to the water box in a 0.15 M NaCl salt concentration to neutralize the simulation systems, and the parameters of the Mg^2+^, Na^+^, and Cl^−^ ions were assigned using the parameters from the works of Joung et al. [87,88].

### 3.2. GaMD Simulations

To improve conformational samplings of the WT and mutated HRAS, three independent GaMD simulations were run on each system. Firstly, each system was subjected to a 30,000-step steepest minimization followed by a 5000-step conjugate gradient minimization to remove bad atom–atom contacts. Subsequently, the temperature was gradually raised from 0 to 310 K for 1 ns using a gentle heating procedure under the NVT condition, wherein heavy atoms were constrained with a weak harmonic constant of 2 kcal·mol^−1^·Å^2^. A temperature of 310 K was adopted to reflect the temperature of the human body. Then, a 2 ns equilibrium process in 310 K was conducted to optimize six systems at the NPT ensemble, and a restriction similar to the heating process was also executed. At last, three conventional molecular dynamics (cMD) simulations were run for 50 ns on six systems at a constant temperature of 310 K and pressure of 1 bar using periodic boundary conditions and the particle mesh Ewald (PME) method [89,90]. Additionally, the initial atomic velocities for each system were assigned with the Maxwell distribution. The last structures from the above three independent cMD simulations were treated as the starting structures for three independent GaMD simulations.

GaMD simulations can reduce the free energy barriers of the systems through harmonic boost potential [43], which efficiently enhances conformational samplings of the WT and mutated HRAS. The duration of each independent GaMD simulation was 1.2 μs. For each system, three independent GaMD trajectories were connected into a single joined GaMD trajectory (SJGT) to facilitate the post-processing analyses. The PyReweighting program [91] proposed by Miao et al. was employed to reweight the data extracted from the post-processing analyses and detect the original free energy of six HRAS-related systems. The SHAKE algorithm constrained all bonds between hydrogen atoms and heavy ones during the cMD and GaMD simulations [92]. The Langevin thermostat with a collision frequency of 2.0 ps^−1^ was adopted to control the temperature [93]. The PME method with an appropriate cutoff value of 10 Å was used to estimate non-bond interactions. The module pmemd.cuda, inlayed in Amber 20, was applied to all simulations [94,95].

### 3.3. GaMD Trajectory-Based Data Process

To investigate the influences of A59E and K117R on conformational transformations of HRAS, root-mean-square deviations (RMSDs) of heavy atoms and root-mean-square fluctuations (RMSFs) of the Cα atoms in HRAS were calculated by wielding the SJGT. Molecular surface areas (MSAs) of the WT and mutated HRAS were also calculated using the SJGT to understand the impacts of A59E and K117R on the structural compactness of HRAS. PCA and calculations of DCCMs [68] were carried out by employing the coordinates of the Cα atoms in HRAS which were recorded in the SJGT, and their details were clarified in our previous work [24]. All of the above analyses were achieved through a program, CPPTRAJ, inlayed in Amber [96]. The FELs were built to provide an energetic basis for understanding conformational changes of the mutated HRAS relative to the WT one using the reaction coordinates recorded in the SJGT through the following equation:(1)$$F(A)=-k_{B}Tlnp(A)$$
in which
(2)$$F\left(A\right)=F^{*}\left(A\right)-\sum_{k=1}^{2}\frac{\beta^{k}}{k!}C_{k}+F_{C}$$
where $F^{*}\left(A\right)=-k_{B}Tlnp^{*}\left(A\right)$ represents the modified free energy arising from GaMD simulations, $F_{C}$ indicates a constant, and ${\beta =k}_{B}T.$ The FELs were built using the PyReweighting program developed by Miao et al., and the details of the reweighting procedure were clarified in the work of Miao et al. [91].

## 4. Conclusions

Oncogenic activity of HRAS caused by point mutations has been observed in cancer patients. Understanding the molecular mechanism of mutation-induced conformational changes in the switch domains is crucial for developing drugs that target the RAS family. We conducted three separate GaMD simulations to improve the sampling of conformational states of the GTP- and GDP-bound WT, A59E, and K117R HRAS. The A59E and K117R mutations greatly affected the switch domains’ structural flexibility in HRAS and significantly influenced their correlated motions and dynamics behavior. These mutations induced conformational changes in both the GTP- and GDP-bound HRAS, particularly impacting the SW1 and SW2 regions. Moreover, the two mutations weakened the hydrogen bonding interactions between GTP/GDP and the switch domains of HRAS and increased the disorder fluctuations of the interactions between Mg^2+^ and the switch domains. Given the pivotal roles of the switch domains in HRAS binding to its effectors, these changes in interaction networks and conformational states are likely to affect the activity and function of HRAS. This study is expected to provide valuable theoretical guidance for drug development targeting the RAS family member.

## Figures and Tables

**Figure 1 molecules-29-00645-f001:**
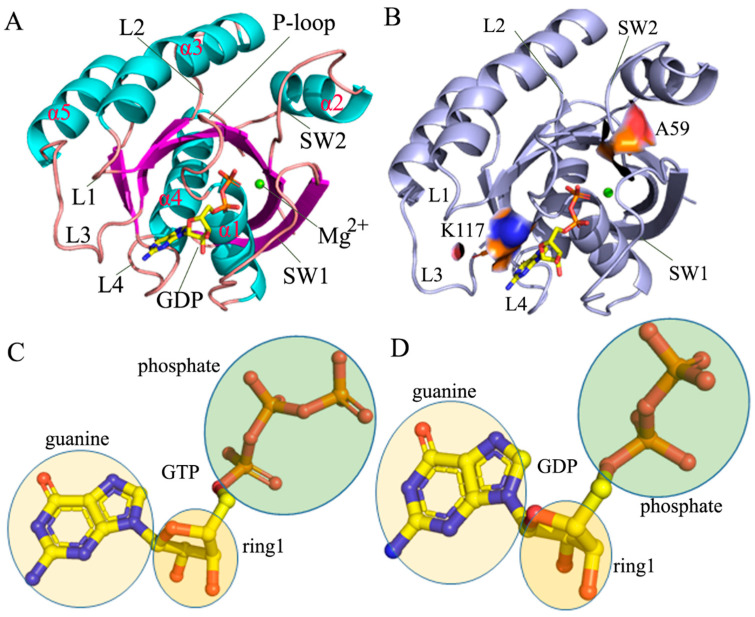
The structures of HRAS, GTP, and GDP. (**A**) The structure of HRAS is shown in cartoon modes with clear labels indicting the second structures. (**B**) Two mutated sites, A59 and K117, are displayed in surface modes. The colors of the surfaces are indicated in red and blue. Structures of GTP (**C**) and GDP (**D**) are characterized by stick modes and colored according to their elements.

**Figure 2 molecules-29-00645-f002:**
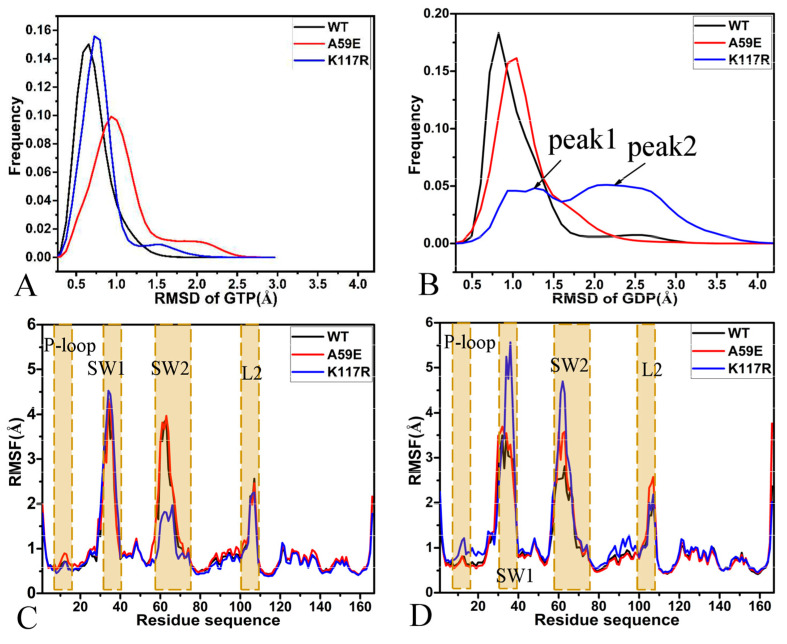
Structural fluctuations and flexibility. RMSDs of heavy atoms for GTP (**A**) and GDP (**B**); RMSFs of the GTP-bound complexes (**C**) and GDP-bound complexes (**D**).

**Figure 3 molecules-29-00645-f003:**
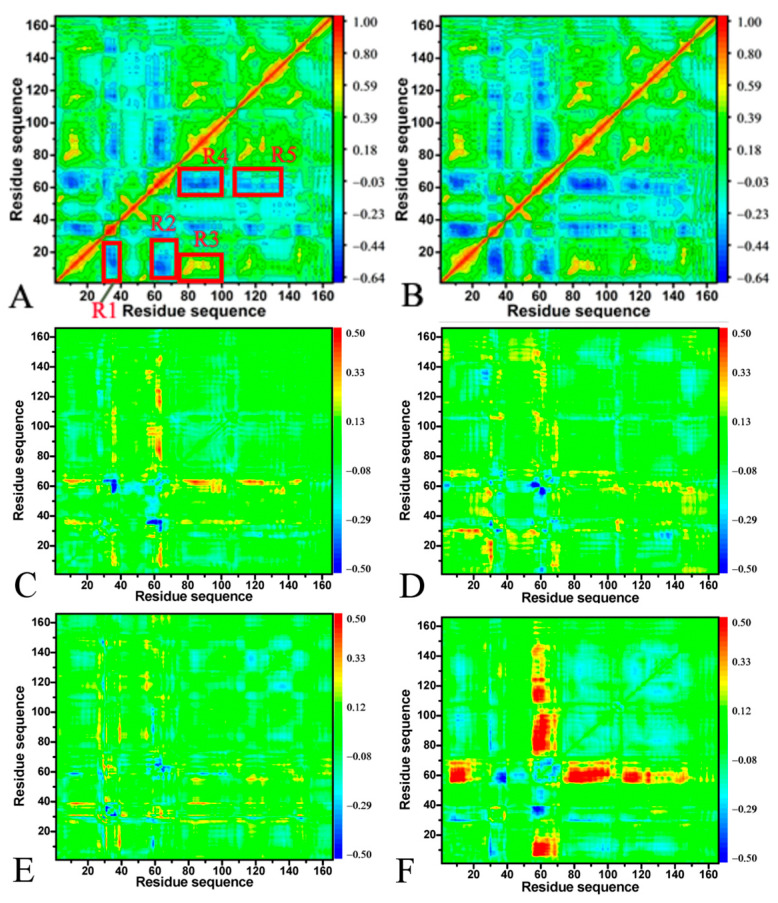
DCCMs calculated using the coordinates of the Cα atoms in HRAS: (**A**) corresponding to the DCCMs of the GTP-bound WT HRAS, and (**B**) corresponding to the DCCMs of the GDP-bound WT HRAS. The differences in DCCMs between the GTP/GDP-bound mutants and the GTP/GDP-bound WT HRAS: (**C**) between GTP-bound A59E and GTP-bound WT HRAS, (**D**) between GDP-bound A59E and GDP-bound WT HRAS, (**E**) between GTP-bound K17R and GTP-bound WT HRAS, and (**F**) between GDP-bound K117R and GDP-bound WT HRAS.

**Figure 4 molecules-29-00645-f004:**
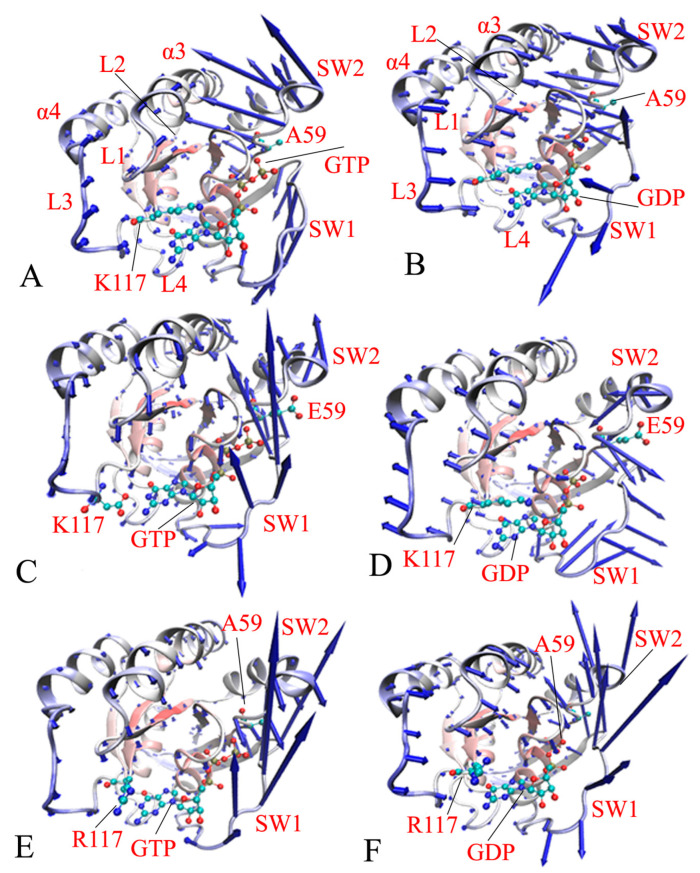
Concerted motions of structural domains from the GTP- and GDP-bound complexes. (**A**) GTP-bound WT, (**B**) GDP-bound WT, (**C**) GTP-bound A59E, (**D**) GDP-bound A59E, (**E**) GTP-bound K117R, and (**F**) GDP-bound K117R, respectively. Arrows on a structure represent the PC vectors.

**Figure 5 molecules-29-00645-f005:**
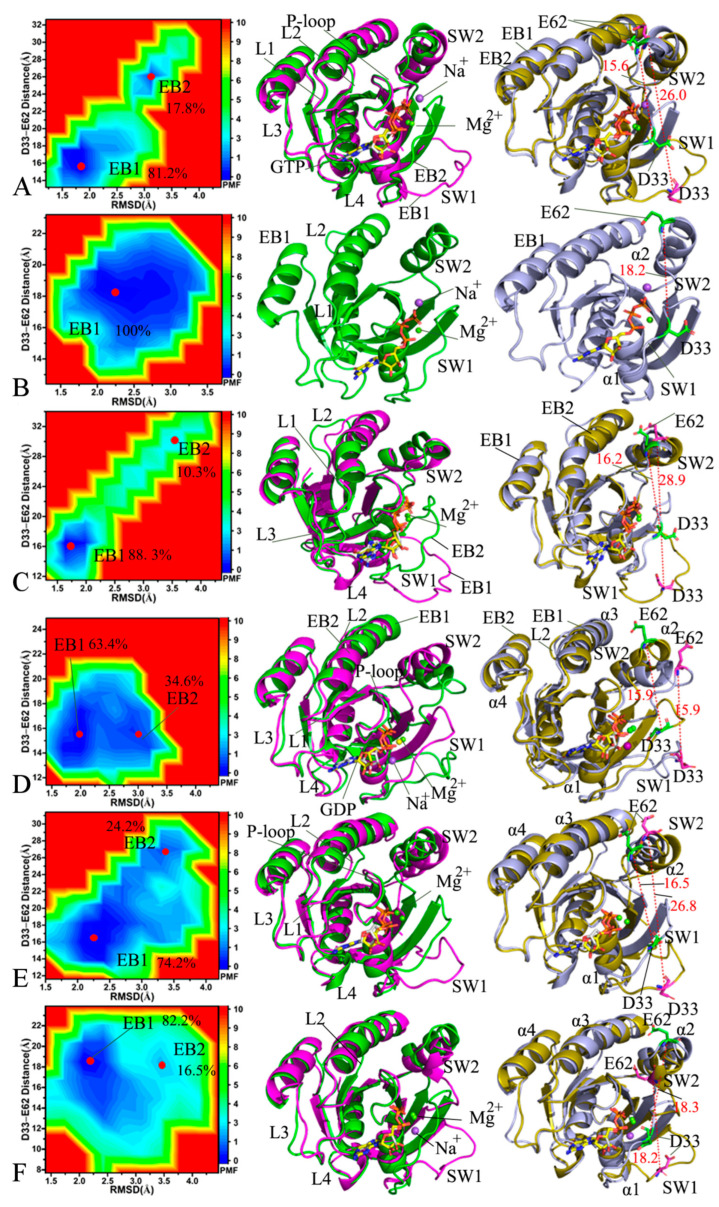
FELs and representative structures of GTP-bound WT mutated HRAS: (**A**) GTP-bound WTHRAS, (**B**) GTP-bound A59EHRAS, (**C**) GTP-bound K117R HRAS, (**D**) GDP-bound WTHRAS, (**E**) GDP-bound A59EHRAS, (**F**) GDP-bound K117R HRAS; the left panels are FELs, the middle panels are the superimposition of two representative structures except for in the GTP-bound A59E complex; the right panels depict relative geometric positions of key residues situated at the SW1 and SW2 from the GTP- and GDP-bound WT, A59E, and K117R HRAS. The PMF is scaled in kcal/mol and the distances are indicated in Å and highlighted in red color.

**Figure 6 molecules-29-00645-f006:**
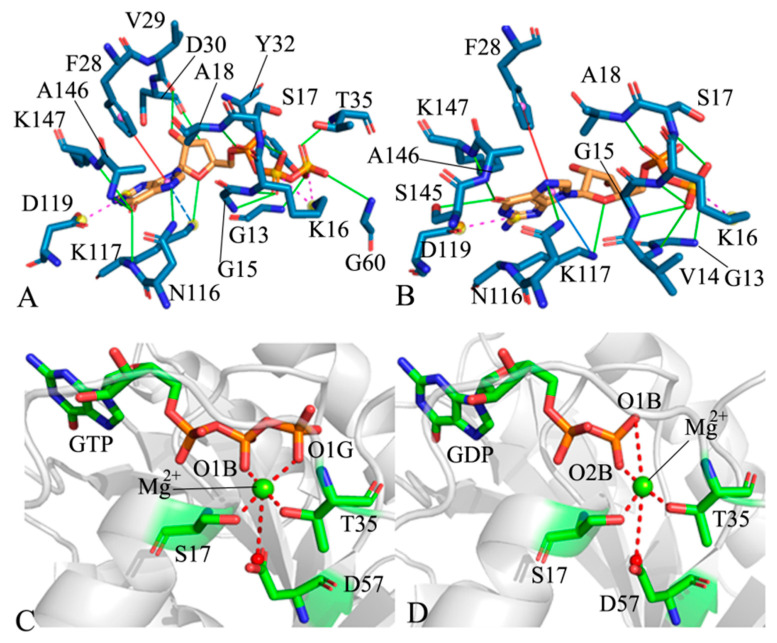
Geometric positions of key interactions. (**A**) GTP–residue interactions, (**B**) GDP–residue interactions, (**C**) the interactions of magnesium ion Mg^2+^ with GTP and residues, and (**D**) the interactions of Mg^2+^ with GDP and residues. The HBIs are shown in solid green lines, salt bridge interactions are shown in dashed purple lines, and the π–π interactions are shown in solid orange lines. Atoms are colored according to their elements.

**Figure 7 molecules-29-00645-f007:**
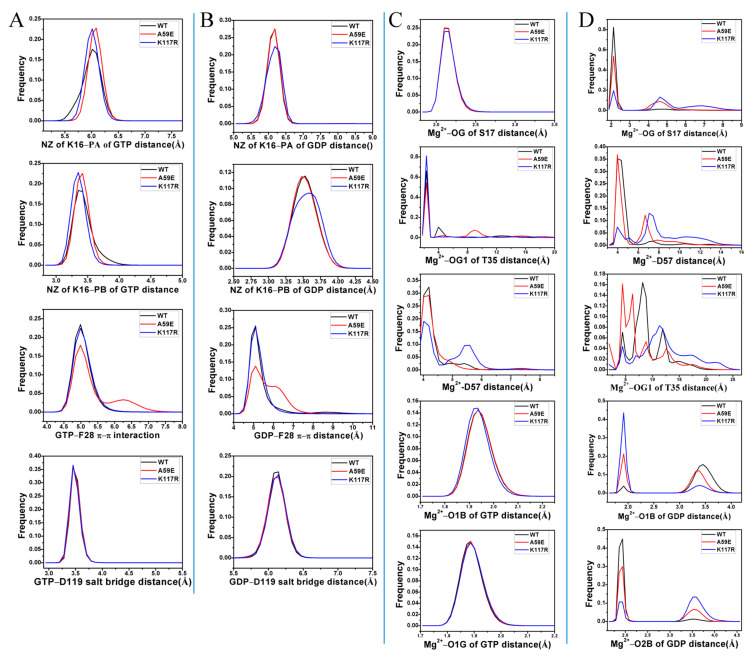
Frequency distribution of distances. The non-bonded interactions of GTP (**A**) and GDP (**B**) with HRAS; the electrostatic interactions of magnesium ion Mg^2+^ with residues and GTP (**C**) and GDP (**D**), respectively.

**Table 1 molecules-29-00645-t001:** Hydrogen bonding interactions of GTP/GDP with the WT and mutated HRAS.

^a^ Hydrogen Bonds		^b^ Occupancy (%)		
Residues	GTP/GDP	WT	A59E	K117R
G13-N-H	O3G/O1B	96.9/96.2	95.2/86.1	85.8/75.4
V14-NH	-/O3B	-/14.2	-/34.3	-/82.1
G15-N-H	O2B/O3B	99.7/96.3	99.6/96.7	98.7/95.4
K16-N-H	O2B/O3B	99.9/99.8	99.9/95.6	99.9/90.8
S17-N-H	O1B/O2B	96.4/62.9	98.4/82.3	92.3/69.1
A18-N-H	O2A/O2A	99.7/94.9	99.7/89.1	99.8/41.2
Y32-N-H	-/O1B	-/29.8	-/5.4	-/1.3
Y32-OH-HH	O3G/-	86.6/-	69.2/-	59.7/-
T35-N-H	O1G/-	65.1/-	45.6/-	42.9/-
V29-O	O2′-HO2′/O2′-H2′	65.4/39.7	49.6/39.7	15.1/9.8
D30-O	O3′-H3T/O3′-H3′	62.3/33.7	50.8/7.4	15.3/9.3
N116-ND2-HD21	N7/N7	95.4/89.7	95.0/88.3	84.1/78.6
K117-NZ-HZ1	O4′/O4′	15.6/18.2	16.0/15.4	-/-
D119-OD1	N1-H1/N1-H1N	99.9/95.4	90.5/84.1	82.6/94.0
D119-OD2	N2-H21/N2-H21	99.9/88.7	89.1/77.5	79.4/89.5
S145-OG-HG	N1/O6	52.8/61.9	53.0/54.3	52.4/63.5
A146-N-H	O6/O6	56.6/56.1	55.3/51.1	67.8/50.5
K147-N-H	O6/O6	87.1/91.7	86.7/82.6	70.2/92.8

^a^ Hydrogen bonds were analyzed with an acceptor·donor distance of <3.5 Å and an acceptor·H-donor angle of >120°. ^b^ Occupancy (%) is defined as the percentage of simulation time for which a specific hydrogen bond existed.

## Data Availability

The data presented in this study are available in article and Appendix A.

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
