# Peer review of "Conformational States of the GDP- and GTP-Bound HRAS Affected by A59E and K117R: An Exploration from Gaussian Accelerated Molecular Dynamics"

_molecules, 2024, doi:10.3390/molecules29030645_

Round 1

Reviewer 1 Report (New Reviewer)

Comments and Suggestions for Authors

In this article the author has described the effect of mutation (A59E and K117R) on the structural fixability of switch domain for GTP- and GDP-bound HRAS. In their study they have reviled that the two mutations lead to the weaker HBIs of GTP/GDP with the switch domains and persuade disordered fluctuations for the interactions between Mg2+ and the switch domains via GaMD simulation. 

The experiments were designed, executed and presented orderly and nicely. The results provided important information about the dynamics as well as interactions of switch domain which is useful for domain/cavity-oriented drug design. I recommended to accept this article as it is. 

Author Response

Thank you very much for this positive comment.

Reviewer 2 Report (New Reviewer)

Comments and Suggestions for Authors

Author Response

Thank you very much for your valuable suggestions and comments that are helpful in revising our manuscript. By following those suggestions and comments, we have revised our manuscript. Our point-to-point responses are accurately incorporated and considered.

Reviewer 3 Report (New Reviewer)

Comments and Suggestions for Authors

The comments are attached

Comments on the Quality of English Language

Grammar and Style must be improved

Author Response

Thank you very much for your valuable suggestions and comments that are helpful in revising our manuscript. By following those suggestions and comments, we have revised our manuscript. Our point-to-point responses are accurately incorporated and considered.

The paper entitled “Conformational states of the GDP- and GTP-bound HRAS affected by A59E and K117R: an exploration from Gaussian accelerated molecular dynamics” presents well organized work on studying the dynamics of HRAS/GDP/GTP conglomerate and focused on two particular point mutations which are related to cancer. From the technical point of view the paper and work presented seem to be more or less correct, the work and data analysis are systematic. Nevertheless I have a strong feeling that this is another technical paper which shows what it might be calculated without a real scientific meaning. Therefore I don’t think the scientific level of the paper is high enough to make it publishable in Molecules.

Reply:Thank you for your comments.

More precisely, why the A59E and / or K117R are important from the cancerogenesis point of view? What epidemiological studies suggest? Are there any convincing experimental studies? How many other mutations were observed, why A99E and K117R were chosen? Why they are important or more important than the others? What we can learn form understanding the mutations effect on the HRAS/GDP/GTP structure? How this knowledge might be applied? Etc etc.

Reply:

In introduction, we clarify why we perform this current study. The revised parts are highlighted in the red and shown below.

“Compared to the frequently detected mutations of codons 12 and 13, A59E and K117R al-so lead to functional difference of HRAS. The study by Ellen et al. proposed that the K117R mutation disrupts guanine nucleotide binding and has similar functional implications as mutations affecting GTP hydrolysis and causing human diseases [27]. On the other hand, A59E is a mutation of RAS associated with cancer. The research conducted by Johnson et al. confirmed that both A59E and phosphorylation significantly accelerate the rate of intrinsic exchange [33].”

In addition, K117R has weaker effects on downstream c-Jun N-terminal kinase signaling.

In our analyses of free energy profiles, the superimposition of representative structures trapped at different energy basins show the effect of mutations on conformational changes. The RMSF analyses and PCA also show that A59E and K117R highly affect conformational dynamics of the switch regions.

Thank you very much for your valuable comment.

Extensive grammar corrections are required through the paper. The same applies to the style.

Reply: The English text has been revised carefully to eliminate any instances of grammatical errors and typographical mistakes.

Reviewer 4 Report (New Reviewer)

Comments and Suggestions for Authors

Zhiping Yu and coauthors used Gaussian accelerated molecular dynamics to analyze the binding of GTP and GDP to wild type HRAS and its two mutants (A59E and K117R). Dynamics of proteins, their free energy landscapes, as well as the correlated motions between structural regions of the GTP- and GDP-bound HRASs, were analyzed.  The manuscript in much part ideologically and structurally repeats the work of Chen et. al.  ( J. Chem. Inf. Model. 2021, 61, 1954−1969).

My remarks concern mainly to the presentation of the results.

All figures and figure captures should be carefully checked and corrected. Some explanations are not sufficient.    

For example, in figure 1 phrase “the colour of the surface is indicated according to elements by default in Pymol,” the phrase may not be clear to all readers. It is better to describe the colors used for surfaces and atoms.

Figure 5.  Did the middle panel show a superposition of representative structures of different energy basins (EB1 and EB2)? How were the portions calculated? For example, the 81.2 and 17.8%. Please describe.  How center of BE2 in Figure 5C was detected?

Some abbreviations, for example, SW1-L, SW1-β, α1-L and so on, have not been described.

In addition, I would recommend that the authors describe the re-weighing procedure in more detail in the Supplementary. I think it will be useful for readers to compare the landscapes of free energy before and after re-weighing and familiarize themselves with technical details of the procedure.

Comments on the Quality of English Language

Minor editing of English language required.

Author Response

Thank you very much for your valuable suggestions and comments that are helpful in revising our manuscript. By following those suggestions and comments, we have revised our manuscript. Our point-to-point responses are accurately incorporated and considered.

This manuscript is a resubmission of an earlier submission. The following is a list of the peer review reports and author responses from that submission.

Round 1

Reviewer 1 Report

Comments and Suggestions for Authors

Line 41: HARS should be HRAS instead.

Lines 45 and 46: unclear and vague.

Line 48: “induced alternations”. What does this expression refer to?

Line 75: the correct verb should be “have been”

Line 114-116: Unclear for which system the authors are describing the results, especially the RMSD ranges. Wt HRAS? The mutants? All of them?

Lines 121-122 : “indicating that A59E and K117R mutations weaken the structural stability of GTP in HRAS compared to the wild-type”. For the K117R mutation, this conclusion seems totally incorrect. The RMSD distribution doesn’t seem to differ in a statistically meaningful way from the black one (a test is needed to confirm statistical significance). Moreover, Fig 2C demonstrates that, if K117R has an effect (see more below), this effect is stabilizing the HRAS fluctuations.

The destabilizing effect of K117R seems more obvious in the case of GDP-HRAS (Fig 2B and 2D). I don’t see it for GTP-HRAS.

Line 139-143: “Differently, A59E slightly enhances the structural flexibility of the switch domain SW2 in the GTP-bound active HRAS, but K117R highly reduces the structural flexibility of this domain compared to the GTP-bound WT one (Figure 2C). In addition, A59E increases the structural flexibility of the P-loop while K117R highly weakens the structural flexibility of the loop L2.”

This paragraph is unsupported by Fig 2. By comparing the black and red lines in panel C, I cannot see any difference at all. The fluctuations of loop L2 are practically unchanged in both mutants, compared to wt (Fig 2C).

When the authors refer to structural flexibility, they should avoid expressions like “highly strengthen the structural flexibility”. Strengthen should be used to mean “more rigid”. Similarly, for the “weakens the structural flexibility”. The authors should replace strengthen and weaken with increase and decrease, where appropriate.

The results in Fig 2 are quite surprising. In Fig 2 panel A, there seems to be no statistically significant difference between the black and blue distributions. However, panel C of Fig 2 shows that the blue line is not identical to the black line. The only substantial difference is in the SW2 region. SW2 is however far from the mutation region. This difference is such that the mutant K117R decreases the fluctuations in SW2. The authors should explain this finding and if it’s observed consistently across multiple simulation replicates.

In Fig 2 panel D, the K117R mutant destabilizes the SW1 and SW2 regions, but again no difference is observed at the mutation site.

Fig 3: I appreciate the DCCM analysis and explicitly plotting the six maps. However, I believe that the readers would be able to understand the differences in the dynamics of the two mutants, with respect to wt HRAS, if the authors replaced maps C and E with the difference map C-A and E-A respectively. Similarly, I suggest replacing D and F with D-B and F-B, respectively. The replaced maps can then be moved to the SI.

Fig 4: please use the orthographic representation in VMD and remove the axes.
The HRAS structures don’t look the same, either because of different orientations or because they are slightly different structures. I can’t understand much about this figure. For an improved clarity, the authors should use just one HRAS structure (the XRAY) for panels A, C and E and only one for panels B, D and F.

Additional important missing pieces of information are the location of (i) the mutation sites and (ii) of the GTP and GDP cofactors.

Fig 5: Why did the author use different ranges for the axes of the FELs in panels A, D and G? With different axes, the comparison is confusing.

In Fig 5E, the authors placed EB2 at coordinates 3.5, 30? Coordinates 3.0, 26 are as good as a choice for EB2, given the pretty identical energy values – within a FEL uncertainty that the authors should calculate and report. I am afraid that the differences between panels A and G are not as substantial as the authors claim to be. The only difference I see is that the state at coordinates 3.5,30 is more favorable in G than in A. The state at coordinates 3.0,26 is more favorable in A than in G. It’s clear that both states are in both wt and mutant HRAS. The authors should therefore: use same axes for the three FELs; use EB2 for state at 3.0,26 and EB3 for state at 3.5,30. Moreover, they should provide the populations of each of these three states in each of the three cases A, D and G.

Fig 6: This figure should be made consistent with Fig 5 (see comments above).

Lines 204-205: if the regions R1 and R2 are already known as SW1 and SW2, the authors should use SW1 and SW2.

Lines 206-209: “Thus the changes in the internal dynamics of the regions R1-R5 caused by A59E and K117R can produce significant impacts on the binding of HRAS to its effectors and allosteric regulation on the activity of HRAS.” Please explain which effects it can produce. If it’s just speculation, please eliminate, as it sounds an unsupported claim.

Fig S1, panel A: are the simulations converged? I see only one transition in the blue time series and one or two at the end of the red time series. The authors should increase the statistics, if the simulations have not converged properly.

Comments on the Quality of English Language

The entire MS contains typos and mistakes. Besides these mistakes, it was very difficult to understand several expressions. They have been highlighted in my feedback.

Reviewer 2 Report

Comments and Suggestions for Authors

Conformational states of the GDP- and GTP-bound HRAS affected by A59E and K117R: an exploration from Gaussian accelerated molecular dynamics

Dear author and editor:

The article could be published after a minor revision. I have some comments on it:

·        The principle role of RAS has to be explained well in the introduction.

·        The discussion part has to be improved.

·        Types of mutation in the RAS and the therapeutic strategy of treatment has to be discussed in a better way.

Thank you, best regards

Reviewer 3 Report

Comments and Suggestions for Authors

Dear Editor,

I read with interest the ms entitled “Conformational states of the GDP- and GTP-bound HRAS affected by A59E and K117R: an exploration from Gaussian accelerated molecular dynamics” (Manuscript ID: molecules-2663182) by Guodong Hu and collaborators, for possible publication on Molecules.

The Authors did an excellent job! In my opinion the topic is of potential interest for the Readership of the journal. The scientific work was conducted in a very professional way, with a solid computational framework. Results are discussed in a robust way and conclusions are well supported by the results. My opinion is therefore that this work should be accepted after minor changes.

. English is poor and a number of grammar mistakes are found. The Authors are strongly recommended to ask for assistance to a native speaker in revising the text. Moreover, a spelling style (either British or American) should be chosen and used consistently thought the whole text.

. Numerous typographic mistakes are also present (e.g. two subsequent full stops, missing brackets, etc.). A careful revision is recommended.

. Whenever a distribution is analysed, the Authors should report not only the position of the maximum (maxima), but also the FWHM or an associated standard deviation.

. The role of the temperature should be emphasised.

. Fig.1(B) – the scale of values associated to the colours of the RDG surfaces must be provided.

. Fig.1(C)&(D) – please provide a legend of colours for the atoms (either in the legend or in the figure panel). The Authors should also say that hydrogen atoms were omitted for the sake of clarity.

. Fig.2 – panels depicting the same quantities should be plot considering the same scales on x and y axes for an easier comparison.  

. I recommend to combine Figs.5 and 6 together. Units should be indicated appropriately.

. Tab.1 – Please replace comma with dot as a decimal separator.

. Figs.8&9 – I recommend to combine them. Please use the same scales on the axes. Please add units where missing.

. Eq.(1) – It should be DeltaGi instead of Gi.

. I recommend to perform quantum mechanical calculations on GTP and GDP molecules and their complexes with Mg(2+) (opt+freq at DFT or MP2 level) and compute the ESP charges and compare geometrical parameters and QM charges with those used in the classical simulations.

Comments on the Quality of English Language

(see report)

Reviewer 4 Report

Comments and Suggestions for Authors

Review Report:

In their manuscript Yu et al. demonstrated a MD simulation study on the oncologic activity of one of the three RAS proteins, HRAS, through point mutations. These mutations lead to conformational alterations in GTP and GDP-bound HRAS, non-covalent forces and disordered interactions between Mg2+ and SW1/2 domains. This is highly important in the context of cancer-related drug discovery. Overall, the manuscript is well written, results are thorough, and fits with the journal scope. I recommend publication of this work essentially as is without any major revision. My specific comments are copied below. I use the following abbreviations, P-page number and L-line number.

Specific Comments:

1. P2-L69: In the figure caption, it should be (B) and not (2).

2. P5-L160: Can the authors provide a total difference in binding energy and/or entropy?

3. P13-L-423: In figure 8, please review the Å unit in x-axis labels.

4. P16-L-542: Please correct the representation of harmonic restriction.